# PinTok: Tokenizers Deserve Dedicated Pinned CPU-Compute and Memory

## Abstract

Tokenization is the first point of contact between large language models (LLMs) and text data, yet it has not been viewed by many as a component of LLMs worth accelerating. During inference, tokenizers typically rely on simple dictionary lookups and are executed on CPUs as standard processes. This approach, however, introduces significant overhead from scheduling delays, core selection, data copying, and other system-level costs. These inefficiencies become problematic in latency-sensitive applications such as embedding, small language models, and agentic AI. In this paper, we present the Pinned Tokenizer (PinTok), a novel tokenizer architecture that reduces redundant hardware, operating system, and networking overhead through three key innovations: core and memory pinning, scheduling and context switch avoidance, and duplicate network packet copy and processing avoidance. Our implementation of PinTok can serve as a drop-in replacement for existing tokenizer deployments, delivering latency reductions of up to 95% (average), 97% (P50), 94% (P90), and 87% (P99) along with throughput improvements of up to 2,084%.

## 1 Introduction

The tokenizer is the neglected child of large language models (LLMs). Tokenization is the first point of contact between LLMs and text data, and while some recent research has highlighted its importance (Zouhar et al., 2023; Ali et al., 2024), there is little prior work focused on understanding, optimizing, and accelerating tokenization at the compute, system, and hardware levels. Today, most tokenizers still operate as generic dictionary-lookup processes running on central processing units (CPUs), and unlike many other components of modern machine learning systems, they are rarely implemented as optimized programs designed for dedicated hardware accelerators.

Table 1: The proportion of wall-clock time spent on the Python tokenizer implementation becomes significant for small language models. Details of the experiments are described in Section 4.

| Model | # Params | Total (ms) | Tokenizer (ms) | Ratio |
|---|---|---|---|---|
| `embeddinggemma-300m` | 308M | 61.4 | 11.9 | 19.4% |
| `ModernBERT-base` | 149M | 40.4 | 15.5 | 38.4% |
| `e5-base-v2` | 109M | 18.3 | 8.8 | 48.0% |

The presumption that the time and resources spent on tokenization are negligible is a **common misconception**, one that we argue has contributed to the lack of system-level research on tokenizers. As shown in Table 1, tokenization can account for a substantial portion of the inference time in smaller models. The need for accelerated tokenizers will only grow as models become more optimized, hardware accelerators improve, model sizes shrink, and applications increasingly depend on ensembles of small language models, such as embedding models, vector databases for Retrieval-Augmented Generation (RAG), and agentic AI systems.

**Contribution.** In this work, we present *PinTok*, a novel fast tokenizer system that reduces previously overlooked sources of overhead arising from computer hardware, operating system, and networking. Specifically, PinTok accelerates tokenizer algorithms through three key techniques: (1)

reducing hardware overhead via core and memory pinning, (2) reducing OS overhead with scheduling and context switch avoidance, and (3) reducing networking overhead with duplicate copy and packet processing avoidance.

We provide a tokenizer implementation on PinTok and evaluate it with multiple models and usage scenarios, demonstrating a significant performance increase. Moreover, PinTok is agnostic to the specific tokenizer algorithm and can be easily used in a drop-in manner, as demonstrated in the following example code comparison.[1]

```python
from transformers import pipeline
pipeline = PythonPipeline("model")
pipeline.start()
... # Retrieve and use tokens
pipeline.stop()
```

```python
from transformers import pipeline
pipeline = PinTokPipeline("model")
pipeline.start()
... # Retrieve and use tokens
pipeline.stop()
```


Standard tokenizer import        PinTok code import


## 1.1 PRIOR WORK

**Improving tokenizer algorithm performance.** Tokenizers for popular models, such as GPT-2 to GPT-4 Radford et al. (2018) and Llama 3 et al. (2024), are built using the Byte-Pair Encoding (BPE) (Gage, 1994) algorithm. However, it is well known that BPE's greedy nature results in inefficient tokenization, especially for tokens that do not appear often in the dataset. To alleviate this issue, algorithms such as WordPiece, adopted by the BERT Devlin et al. (2019) tokenizer, and SentencePiece Kudo & Richardson (2018), adopted by the XLM tokenizer Lample & Conneau (2019), emerged, each of which promises to solve problems such as inefficient merging or splitting of words. Then, there are follow-up works that optimize the complexity of these methods by providing improved algorithms. Some notable examples include Fast WordPiece Tokenization (Song et al., 2021) and Sennrich et al. (2016). Finally, a more recent promising work introduces transformers to dynamically tokenize incoming data (Pagnoni et al., 2025). PinTok is an architecture design that can complement all listed works to provide additional performance and efficiency gains.

**Using existing ML accelerators for tokenizers.** There are works that perform tokenization on popular AI accelerators, such as graphical processing units (GPUs). Some examples include Parallel BPE (You, 2025) and RAPIDS cuDF tokenizer system (Jawa, 2021). However, AI accelerators excel at large batches of parallel floating point operations, whereas most tokenizer algorithms consist of repetitive byte manipulations and dictionary look-ups, thus these approaches have not shown reasonable latency benefits. Due to such reasons, popular models such as BERT (Devlin et al., 2019), GPT-OSS (Radford et al., 2018), Llama (et al., 2024), and Qwen (Yang et al., 2025) all utilize tokenizers provided by frameworks such as HuggingFace that run as a standard Python, C, or Rust program on a CPU. Unlike such works, PinTok does not require hardware changes and can be easily run on general-purpose machines.

**Acceleration via network offload.** Accelerating applications using network software or hardware, an idea similar to the use of GPUs for AI, is not entirely new (Kianpisheh & Taleb, 2023). Recent advances in network offload began with the advent of frameworks such as the Data Plane Development Kit (DPDK) (Foundation, 2015) and eBPF (Borkmann et al., 2024), and programmable network hardware such as Tofino (Keslassy et al., 2018) and Nvidia (NVIDIA Corporation, 2025) Bluefield NICs. Using such frameworks, early research on programmable network offload focus on improving simpler applications, such as caches, databases, and consensus protocols (Jin et al., 2017; Psaras et al., 2014; Dang et al., 2020). Follow-up research on advanced applications, such as lambda functions and blockchain, have also shown significant benefits from network offload. Lastly, using network offload to improve ML workloads, there are relevant work on training (Lao et al., 2021; Sapio et al., 2021), inference (Sanvito et al., 2018; Zheng et al., 2022; Xiong & Zilberman, 2019; Kapoor et al., 2025), and data processing (Sapio et al., 2017b). *PinTok* is a form of network offload for data processing, but no work has specifically focused on utilizing network offload for tokenizers.

---

[1] PinTok requires a one-time installation of a boot configuration. Details of the configuration and installation instructions can be found in Appendix A.

Table 2: System-level overheads commonly incurred by conventional tokenizers. These inefficiencies are mitigated in PinTok.

| Sources of overhead | Type | Latency |
|---|---|---|
| Core selection
TLB miss & no page swap
Page swap | **Hardware** | $\sim 1\mu s$ to $20\mu s$
$\sim 1\mu s$ to $10\mu s$
$\sim 50\mu s$ to 10ms |
| Process context switching
Process scheduling | **Operating system** | $\sim 2\mu s$ to $20\mu s$
$\sim 1\mu s$ to 20ms |
| Packet copy to kernel
Network packet interrupt
Kernel network stack packet processing | **Networking** | $\sim 0.5\mu s$ to $2\mu s$ per packet
$\sim 1\mu s$ to $5\mu s$ per packet
$\sim 10\mu s$ to $50\mu s$ per packet |

## 2 COMPUTER SYSTEM PRELIMINARIES

In this section, we provide a brief introduction to the basic concepts of computer systems that are relevant to this work and outline a list of potential overheads that PinTok aims to mitigate. The discussion is intended for broader machine learning audiences that may not be familiar with the lower-level systems considerations.

**Hardware: Memory, CPU, and disk.** *Memory* (RAM) is a high-speed storage medium with a physical and a virtual address. Modern operating systems manage memory by dividing it into fixed-size blocks called *pages*. Pages can also be configured as *superpages*, which are larger page that improve memory efficiency (Navarro et al., 2002).

A *central processing unit (CPU)* is responsible for executing software instructions and consists of one or more cores, each of which can operate independently. The CPU also handles memory addressing, i.e., translating between physical and virtual memory addresses, with the assistance of the memory management unit (MMU) and the translation lookaside buffer (TLB). The MMU performs address translation by referring to the page table, while the TLB is a hardware cache designed to accelerate this translation. A *TLB hit* occurs when the required translation is found in the TLB, whereas a *TLB miss* occurs when it is not. Superpages improve the efficiency of TLB operations by reducing the number of required translations. The CPU is also responsible for servicing interrupts generated by hardware devices such as the network interface card (NIC). When an interrupt is triggered, the CPU core currently executing a task is paused, and an interrupt service routine (ISR) is invoked to handle the event. It is possible to dedicate a CPU core to a specific process and exclude it from interrupt handling to achieve more predictable performance for that process.

*Disk* is a slower, non-volatile storage medium often used during context switching, which saves the state of a running process so it can resume later and frees memory for new processes. When a process is swapped out, its data in memory is written to disk in an operation called *page swap*, a slow process that introduces latency. To reduce page swaps, systems often enable superpages, since larger pages are less likely to be swapped.

**Operating system: Resource and process management.** The operating system (OS) is the software that manages a computer's hardware and software resources. At its core is the kernel, which contains the critical logic for system management. To ensure security, the OS divides memory into user space and kernel space. User space is a less privileged region where applications run, while kernel space is a highly privileged area for the kernel, drivers, and extensions. Two of the OS's main responsibilities are resource management and process management.

*Resource management* involves allocating and sharing hardware resources, such as the CPU and memory, among multiple processes running on a system. The CPU is shared by assigning time slices of its cycles to different processes. Memory is shared by dividing available capacity among processes and performing page swaps when necessary to free up space.

*Process management* involves scheduling, executing, and terminating processes while providing the resources they require. When a process starts, it first enters a scheduling queue. The process scheduler then uses a scheduling algorithm to select the next process to run. Once selected, the OS determines which core will execute the process. After a core is chosen, the system saves or clears the existing context on that core and then loads the new process's context, including its instructions and variables. At this point, the process begins executing its instructions until it terminates.

**Networking: Processing incoming packets.**   Once data packets arrive at the host, they are first received and queued by the network interface card (NIC). The NIC then sends an interrupt to the CPU, signaling it to process the packets when ready. When the CPU begins processing, it runs an interrupt service routine (ISR) to copy the packets from the NIC's memory into the OS's kernel space. The kernel's networking stack then performs protocol operations, such as TCP/IP header processing, and routes the packets to the appropriate applications.

Because this process involves multiple data copies, a kernel-bypass approach can be used to transfer packets directly from the NIC to user space. Some software further reduces latency by polling for incoming packets instead of waiting for NIC interrupts, enabling immediate processing as packets arrive. Taking this further, certain applications integrate custom protocol logic to avoid redundant packet processing. Frameworks such as the Data Plane Development Kit (DPDK) (Foundation, 2015) enable these techniques, allowing applications to process network data with significantly lower latency.

**Hardware overhead.**   The two main sources of overhead in memory access are TLB misses and page faults, which are attempts to access data that are currently not in physical memory. This overhead can be modeled as the effective access time:

$$T_{\text{hardware}} = (1 - m) \times (T_{\text{tlb}} + T_{\text{mem}}) + m \times (T_{\text{tlb}} + T_{\text{ptw}} + (1 - p) \times T_{\text{mem}} + p \times T_{\text{pf}}),$$

where $m$ is the probability of a TLB miss, $T_{\text{tlb}}$ is the time to access the TLB, $p$ is the probability of a page fault, $T_{\text{mem}}$ is the time to access the main memory, $T_{\text{ptw}}$ is the time for a page table walk, and $T_{\text{pf}}$ is the time to service a page fault. Often, $T_{\text{pf}}$ represents the most significant cost. PinTok's goal is to reduce both $m$ and $p$ in order to bring the effective $T_{\text{hardware}}$ closer to the theoretical minimum of $(T_{\text{tlb}} + T_{\text{mem}})$, and in particular to minimize the contribution of the $mp \times T_{\text{pf}}$ term.

**Operating system overhead.**   The three primary sources of OS scheduling overhead are *core selection*, *process context switching*, and *scheduler execution*. This overhead can be modeled as

$$T_{\text{OS}} = T_{\text{cs}} + T_{\text{pcs}} + T_{\text{exec}},$$

where $T_{\text{cs}}$, $T_{\text{pcs}}$, and $T_{\text{exec}}$ respectively correspond to the three overhead sources. PinTok aims to eliminate $T_{\text{cs}}$ and $T_{\text{pcs}}$ through core pinning and interrupt avoidance.

**Networking overhead.**   The networking overhead is given by

$$T_{\text{networking}} = T_{\text{interrupt}} + T_{\text{copy}} + T_{\text{protocol}},$$

where $T_{\text{interrupt}}$ is the time to perform interrupt handling, $T_{\text{copy}}$ is the time to copy packets into kernel memory, $T_{\text{protocol}}$ is the time to perform protocol logic. PinTok aims to eliminate $T_{\text{interrupt}}$ via polling, reduce $T_{\text{copy}}$ via kernel-bypassing and reduce redundant processing by merging tokenizer algorithm time with $T_{\text{protocol}}$.

## 3   PINTOK TOKENIZER SYSTEM

The core techniques behind PinTok are: (1) reducing hardware overhead through core and memory pinning, (2) minimizing OS overhead by avoiding scheduling and context switches, and (3) lowering networking overhead through copy avoidance and eliminating duplicate packet processing. This section explains how PinTok realizes these techniques, and Figure 1 illustrates the key ideas. For a detailed comparison, Algorithm 1 presents a conventional tokenizer system, while Algorithm 2 shows the optimized PinTok.

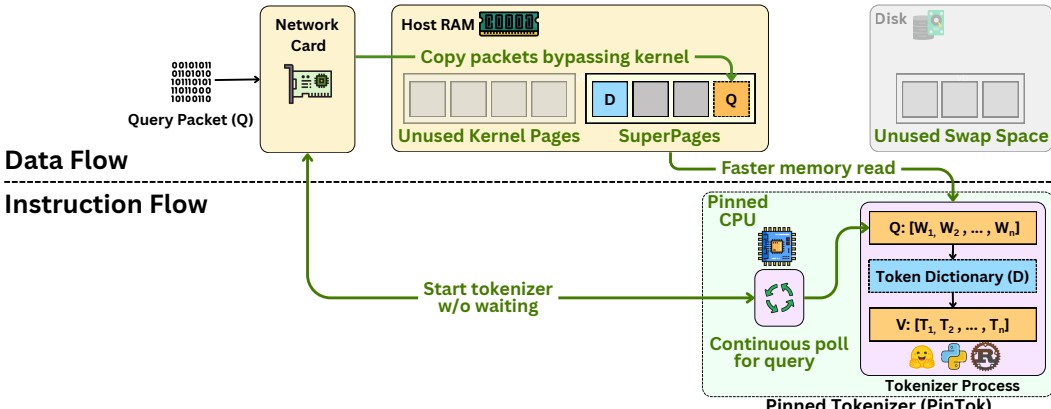

(a) Overview of the tokenizer flow on a conventional Python/Rust tokenizer provided by frameworks like HuggingFace. The red dotted lines denote the potential sources of overheads, which include the overheads mentioned in Section 2.

(b) Overview of tokenizer flow on PinTok where the sources of overheads are addressed.

Figure 1: Comparison of the tokenizer flow between conventional tokenizers vs PinTok.

**Reducing hardware overhead: Core and memory pinning.** PinTok implements core pinning by dedicating a set of CPU cores to itself at boot time, ensuring that no other processes, including hardware interrupts, can use those cores. This eliminates the overhead and nondeterminism of runtime core selection and improves tokenizer performance, particularly in tail events. PinTok also implements memory pinning by reserving a large, fixed, unswappable page, which prevents scheduler-induced interruptions by removing the possibility of page swaps and improves TLB hit rates since a single TLB entry can map the entire page. In practice, these techniques are implemented through OS boot configurations. Details on enabling such settings are provided in Appendix A.

**Reducing OS overhead: Scheduling and context switch avoidance.** PinTok avoids scheduling and context switching by keeping the tokenizer process running continuously and dedicating a set of CPU cores exclusively to tokenizers. It operates in a continuous poll mode, continually consuming CPU cycles to request new queries for tokenization. This reduces the likelihood of being context-switched, since the OS tends to switch out processes that are not actively utilizing the CPU. Furthermore, PinTok prevents dedicated cores from being allocated to kernel processes by enabling kernel bypass to retrieve network packets directly from the NIC. An additional benefit of this design is that tokenizers execute in a run-to-completion (RTC) manner, ensuring that once a query is received, it is processed without interruption. As a result, tokenizer performance becomes significantly more predictable, enabling tighter service-level guarantees.

**Algorithm 1** Conventional tokenizer system

**Require:** $\mathbf{P} = \{p_1, p_2, \ldots, p_N\}$
 ▷ Each $p_i$ contains a fraction of the input query
 **procedure** KERNELNETWORK($\mathbf{P}$)
  **for** each $p_i$ in $\mathbf{P}$ **do**
   NICINTERRUPT(CPU)
   STARTISR(CPU)
   $p_{k_i} \leftarrow$ COPYTOKERNEL($p_i$)
    ▷ Extra copy into kernel memory
   $p_{u_i} \leftarrow$ COPYTOUSERSPACE($p_{k_i}$)
    ▷ Extra copy into user-space memory
   ENQUEUE(queue, $p_{u_i}$)
**Require:** $N$ (number of fragments for the next query)
 **procedure** TOKENIZER(queue)
  WAITUNTIL($\big|\{\text{seq}(p) : p \in \text{queue}\}\big| \geq N$)
  $\mathbf{B} \leftarrow$ DEQUEUEUNTIL(queue, $N$)
  $q \leftarrow$ REASSEMBLEANDORDER($\mathbf{B}$)
  $\mathbf{T} \leftarrow$ TOKENIZE($q$)
   ▷ $\mathbf{T}$ is a set of token IDs.

**Algorithm 2** PinTok optimized system

**Require:** Token IDs to Python stage *without copies*
**Ensure:** Zero-copy handoff of $\mathbf{T}$ to Python
 PIN(*cores* := dedicated RX/token cores)
 PIN(*mem* := superpages/pinned pools)
 INITNIC(RX queues bound to pinned mbufs)
  ▷ Preallocate token-ID buffer for Python
 $mz \leftarrow$ ALLOCPINNEDMEMZONE
 **procedure** PINTOK
  **for** each $p_i$ from NICSTREAM **do**
   $\mathbf{T}_{\text{range}} \leftarrow$ TOKENIZE($p_i \rightarrow mz$)
   PUBLISHTOPYTHON($mz$, $\mathbf{T}_{\text{range}}$)
    ▷ **zero-copy** view
   RECYCLERXBUF($p_i$)

*The descriptions of each function are provided in Appendix C.

**Reducing networking overhead: Duplicate copy and packet processing avoidance** PinTok avoids redundant network packet copying and processing by bypassing the kernel to retrieve packets directly in the user space in a poll-based manner, and performing tokenization along with the network protocol logic within the same user space program to reduce multiple packet processing. Conceptually, this approach is analogous to kernel fusion in deep learning: instead of launching multiple kernels and incurring synchronization overhead at each boundary, operations are fused into a single execution. In a conventional network data path, packets are first buffered in kernel pages, then processed by kernel-level protocol logic to identify the target application, and finally copied into application memory for further processing. Each of these stages introduces additional latency, consumes CPU cycles, and increases memory bandwidth usage, all of which PinTok avoids.

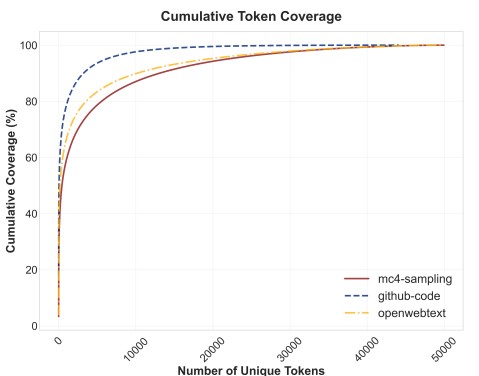

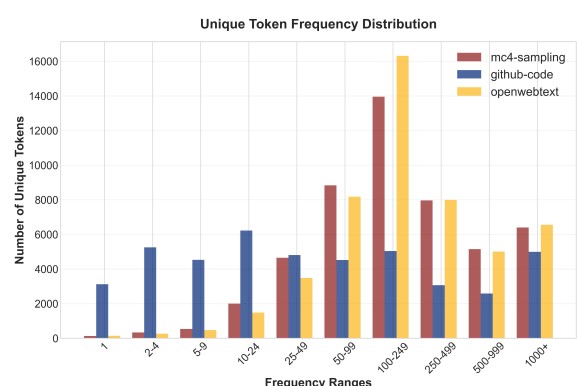

(a) Cumulative token coverage that shows the number of tokens required to cover the portion of the entire data set. We can see that few unique tokens dominate the github-code dataset, whereas natural language is more diverse.

(b) The number of unique tokens is distributed across different frequency ranges. For example, in the frequency range of 1, the GitHub-code dataset shows many tokens that appear only once, whereas such single-occurrence tokens are much less common in natural language datasets.

Figure 2: Token distributional characteristics across datasets.

## 4 EXPERIMENTS

We evaluated PinTok both as a standalone tokenizer and as part of end-to-end pipelines. All experiments were run on a dedicated Kernel-based Virtual Machine (KVM) (see Appendix B for details). To ensure diverse evaluation, we used three representative corpora spanning different domains: `openwebtext` (English web text, Gokaslan et al. (2019)),

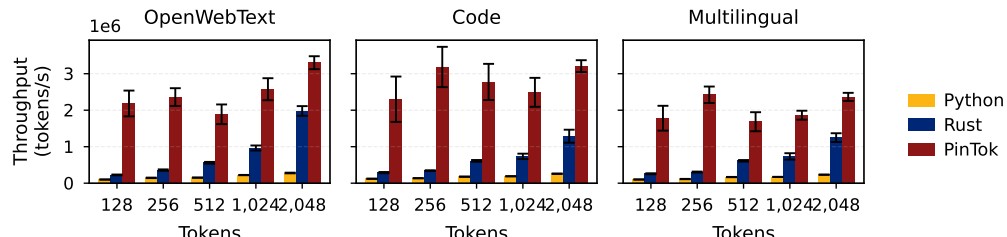

Figure 3: Tokenizer throughput across input lengths (128-2048 tokens per input) on three datasets. Higher is better. PinTok consistently achieves higher throughput than Rust and Python baselines.

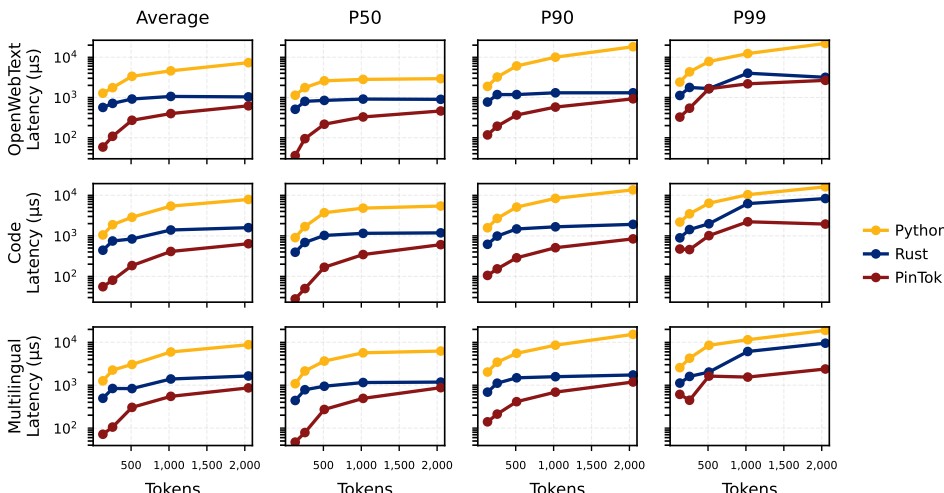

Figure 4: Latency distributions (average, P50, P90, P99) across datasets and input lengths (128-2048 tokens per input). (Lower is better.) PinTok consistently achieves lower latency than Rust and Python baselines.

`bertin-project/mc4-sampling` (multilingual web text, la Rosa et al. (2022)), and `codeparrot/github-code` (source code, CodeParrot Team (2022)). The multilingual corpus included documents in five languages (English, French, German, Spanish, and Chinese) to test cross-lingual performance. This mix covers typical English web content, diverse languages, and structured programming text, testing PinTok on a wide range of token distributions.

We benchmarked PinTok's tokenization throughput and latency on these corpora to quantify performance gains. For comparability, we included a re-implementation of the GPT-2 byte-level BPE tokenizer with its original vocabulary and merge rules (Radford et al., 2019) as a baseline. We also evaluated PinTok in complete embedding pipelines by pairing it with three different embedding models: *ModernBERT-base*, an 8K-token BERT encoder (Warner et al., 2025), *E5-base-v2*, a 12-layer contrastive model (Wang et al., 2024), and *EmbeddingGemma-300m*, a 300M-parameter multilingual embedding model (Choi et al., 2025). These models range from a widely-used BERT variant to a newly released embedding model, demonstrating that PinTok generalizes across tokenization algorithms and integrates seamlessly with both established and emerging architectures.

Beyond domain and language diversity, the chosen corpora exhibit markedly different token frequency profiles. Figure 2 illustrates these differences: in panel (a), the source code corpus achieves coverage saturation with far fewer unique tokens (reflecting its repetitive structure), whereas the natural language corpora require many more unique tokens for the same coverage. In panel (b), the code dataset contains a larger fraction of rare tokens, while the web text corpora have more mid-frequency tokens. Evaluating on such diverse corpora ensures a rigorous stress-test of PinTok across varying domains, languages, and token distributions, highlighting its efficiency and broad applicability.

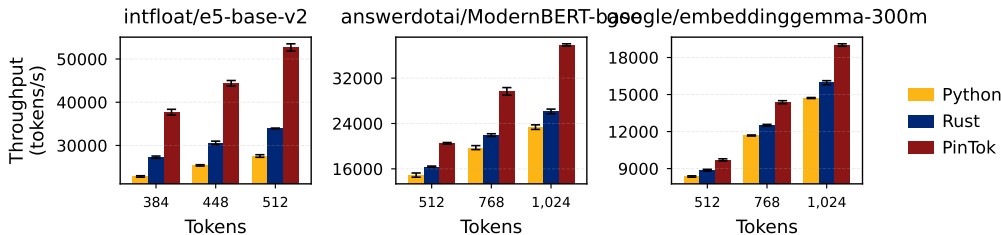

Figure 5: Embedding throughput with three representative models (`e5-base-v2`, `ModernBERT-base`, `embeddinggemma-300m`) on varying input lengths (128-2048 tokens per input). (Higher is better). PinTok consistently achieves higher throughput than Rust and Python baselines.

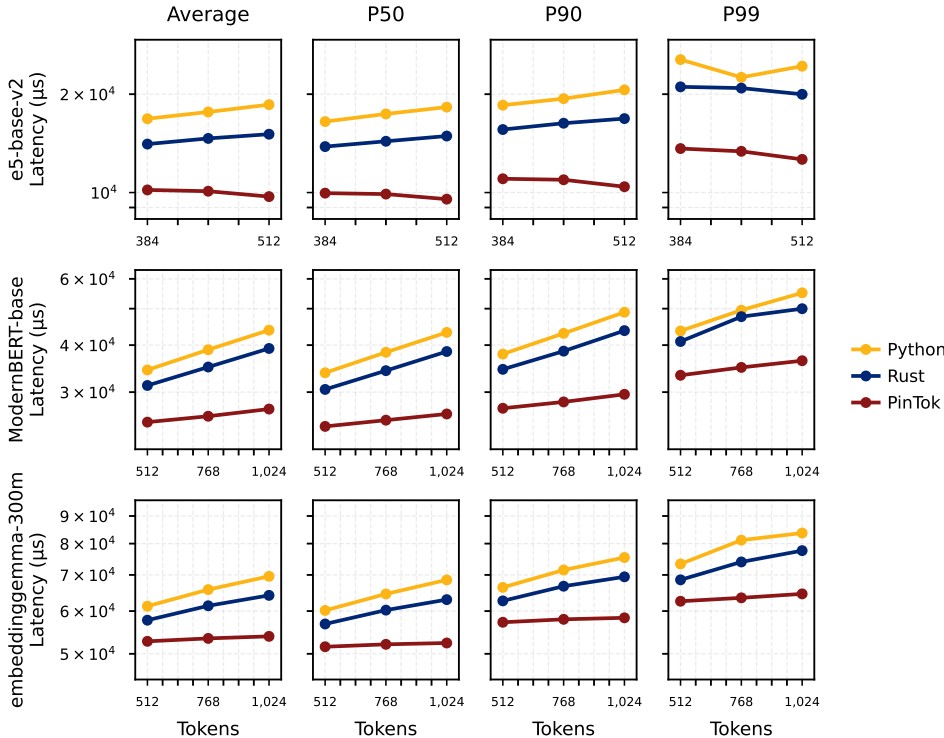

Figure 6: Latency distributions (average, P50, P90, P99) for embedding pipelines for three models on varying input length (128-2048 tokens per input). (Lower is better.) PinTok consistently achieves lower latency than Rust and Python baselines.

## 4.1 TOKENIZER BENCHMARK RESULTS

We benchmarked the throughput and latency of PinTok against a high-performance Rust tokenizer and a Python baseline. All results were averaged over 20 independent trials. Figure 3 shows throughput (tokens per second) as a function of input size (128–2048 tokens), and Figure 4 reports the tokenization latency distribution for each dataset, showing the average (mean) latency as well as the median (P50), 90th percentile (P90), and 99th percentile (P99) tail latencies. PinTok consistently improves the throughput and latency. For completeness, the detailed numerical results for all datasets and configurations are provided in Appendix D.

## 4.2 END-TO-END EMBEDDING PIPELINE RESULTS

To assess downstream impact, we paired PinTok, Rust, and Python tokenizers with three embedding models: `e5-base-v2`, `ModernBERT-base`, and `embeddinggemma-300m`. We measured

throughput and latency for inputs of 512–1024 tokens for `ModernBERT-base` and 384–512 tokens for `e5-base-v2`, averaged over 5 trials. Figure 5 shows the throughput, and Figure 6 reports the latency. These results show that PinTok's optimizations carry through the full embedding pipeline, improving both speed and predictability. For completeness, the detailed numerical results for all datasets and configurations are provided in Appendix E.

### 4.3 END-TO-END ML APPLICATIONS RESULTS

We also evaluate the significance of the end-to-end speedups provided by PinTok in several application domains. The first and second applications is the retrieval and insertion times in a popular open-source vector database, ChromaDB (contributors, 2025), which is widely used to support RAG use cases. The third application focuses on querying a small language model, Qwen3 4B (Yang et al., 2025), which is commonly employed as a classifier in the classical machine learning sense or as a router in agentic AI applications (Wu et al., 2025). The fourth application involves querying a large language model (LLM), GPT-OSS 120b (OpenAI, 2025), which is often used for more comprehensive question-answering and reasoning tasks. For the first and second workflows, we report the total time required to complete the operations, whereas for the third and fourth workflows, we report the Time-to-First-Token (TTFT), due to the variable nature of the output depending on the input and infrastructure configuration.

Table 3: Average latency ($ms$) and the latency saving of PinTok with respect to Python for various ML applications and models. ChromaDB retrieval and insertion are measured using the experimental setup of Appendix B, whereas SLM and LLM TTFTs are obtained on an H100 GPU.

| Application | PinTok | Rust | Python | Latency Saving (%) |
|---|---|---|---|---|
| ChromaDB retrieval | 11.35 | 16.73 | 20.06 | 43% |
| ChromaDB insertion | 47.55 | 52.93 | 56.26 | 16% |
| Qwen3 4B query (TTFT) | 413.55 | 418.93 | 422.26 | 2.1% |
| GPT-OSS 120b query (TTFT) | 1348.55 | 1353.93 | 1357.26 | 0.64% |

The results in Table 3 clearly demonstrate that latency reductions are more pronounced for shorter workflows, as the improvements from PinTok primarily stem from the tokenizer. However, even for large language models, a latency reduction of approximately 1% can translate into a 1% improvement in overall system efficiency. We argue that, at scale, such improvements can be highly significant, especially given PinTok is a software solution and does not require new hardware.

## 5 CONCLUSION

In this work, we present PinTok, a tokenizer system that achieves acceleration through system-level optimizations. We provide experimental results demonstrating the significance of the speedup provided by PinTok.

Our work highlights how system-level treatment of tokenization can unlock both research and production gains. A promising direction is to adapt subword algorithms for streaming so they integrate more naturally with PinTok. In byte-pair encoding (e.g., GPT-2's byte-level BPE), tokenization repeatedly applies ranked pair-merges; in WordPiece (e.g., BERT), tokenization performs greedy longest-match segmentation over a learned subword vocabulary. In both cases, the correct segmentation can depend on context that spans chunk or packet boundaries. When run strictly per-packet, the tokenizer may commit early and yield segmentations that are suboptimal or that diverge from the full-context (offline) tokenizer's output. Investigating streaming-aware adaptations that preserve accuracy while enabling low-latency execution is therefore a valuable avenue for future work.

Another avenue for future work is implementing PinTok directly at the hardware level, specifically on the NIC. In-network computing is an emerging research area that leverages the compute capabilities of NIC (instead of CPU or GPU compute) for processing streaming data, with the primary goal of reducing latency (Sapio et al., 2017a). Deploying PinTok on NICs presents a significant potential for further efficiency gains.

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

## A  PINTOK INITIALIZATION CONFIGURATION

PinTok requires a one-time modification of the boot configuration. For a more portable deployment of PinTok, these configurations can easily be baked into an OS image (e.g., Amazon Machine Images) so that the configurations are done seamlessly.

### A.1  CORE PINNING

Core pinning requires few configurations in detail. First, `isolcpus` removes the specific CPU from being considered for task scheduling. Second, `nohz_full` make sure that the kernel tries to move away from that CPU as much kernel noise as it can Third, we need to remove the CPU from participating in Read-Copy-Update (RCU). RCU system is a lockless mechanism for mutual exclusion inside the kernel, so as a consequence of performing RCU operations, call-backs are sometimes queued on CPUs to be performed at a future moment when removing memory is safe. We can remove this by setting `rcu_nocbs`. Fourth, we need to remove a CPU from servicing CPU interrupts by setting the `irqaffinity` parameters. All of this can be done by setting the GRUB parameter as follows.

```
GRUB_CMDLINE_LINUX="... isolcpus=0 nohz_full=0 \
    rcu_nocbs=0 irqaffinity=1-"
```

Finally, we need to ensure that the CPU is running at the fastest clock by updating the setting on the CPU governor to performance. This can be done by running the following.

```
sudo cpupower frequency-set -g performance
echo 'GOVERNOR="performance"' | sudo tee /etc/default/cpupower >/dev/null
sudo systemctl enable --now cpupower
```

### A.2  MEMORY PINNING

The main idea behind memory pinning is to enable superpages, which is a way to increase each page size, so that the pages are not swapped out and TLB efficiency is greatly increased (since each TLB entry maps to a much larger memory space). Once superpages are created, they can be dedicated to a specific application, enabling the memory pinning effect to take place. The superpages can be enabled by modifying the boot parameter as follows.

```
# /etc/default/grub
GRUB_CMDLINE_LINUX="... default_hugepagesz=1G hugepagesz=1G hugepages=8"
```

This enables 8 superpages of size 1G, occupying 8GB of RAM in total. The configuration can be easily modified based on the type of system and constraints. Once created, you must mount the superpages as follows.

```
nodev /mnt/huge hugetlbfs pagesize=1GB 0 0
```

Then the superpages are ready to use for the tokenizers by using the mount point as the memory location.

### A.3  DPDK

For network level optimization, we utilized DPDK (Foundation, 2015). The DPDK installation instructions are highly fluid based on the software version, thus we provide the URL (`https://doc.dpdk.org/guides-<version>/linux_gsg/index.html`) to the installation manual to select the version of DPDK of interest. Similarly to other configurations, it is trivial to enable DPDK as part of the OS installation. In fact, DPDK is widely used in data center settings to deploy fast networking logic to hundreds of thousands of servers and users maybe using DPDK without them knowing. Therefore, we believe that PinTok can follow the same deployment patterns without problems.

## B    EXPERIMENTAL SETUP

All experiments were conducted on a Linux host running Ubuntu 24.04 with an **AMD Ryzen 7 2700** eight-core processor (3.2 GHz base frequency), **32 GB DDR4 RAM**, and an **NVIDIA GeForce RTX 4090 GPU**. The GPU was configured with the `nvidia-driver-535.247.01` (CUDA 12.2).

To enable portability across heterogeneous hardware, we deployed PinTok inside a **KVM-based virtual machine**. The VM was allocated **4 vCPUs** (with one core dedicated and pinned to the DPDK datapath), **24 GB RAM**, and full GPU passthrough via the `vfio-pci` driver. This configuration ensured that the guest environment could directly access the RTX 4090 while preserving isolation and reproducibility across hosts.

Within this virtualized environment, packets were transmitted to a designated port and ingested by DPDK for tokenization. Details of **core pinning**, **memory pinning**, and **DPDK initialization** are provided separately in Appendix A. The PinTok codebase used in these experiments will be made publicly available to support reproducibility.

## C    FUNCTION DESCRIPTION FOR ALGORITHM 1 AND ALGORITHM 2

In order to aid in understanding of Algorithm 1 and Algorithm 2, this section provides a high-level description of each helper function.

**Conventional tokenizer system**

- `KERNELNETWORK(P)`: Top-level routine that receives packets from the NIC and forwards them through kernel memory, user-space memory, and finally to the tokenizer.

- `NICINTERRUPT(CPU)`: Triggers a hardware interrupt to notify the CPU that new packets have arrived.

- `STARTISR(CPU)`: Begins the interrupt service routine (ISR) on the CPU to process incoming packets.

- `COPYTOKERNEL(p_i)`: Copies packet $p_i$ from the NIC into kernel memory (first extra copy).

- `COPYTOUSERSPACE(p_{k_i})`: Copies the kernel-space packet $p_{k_i}$ into user-space memory (second extra copy).

- `REASSEMBLEANDORDER(B)`: Reconstructs the original query $q$ from the packet fragments in $B$.

**PinTok optimized system**

- `PIN(cores := dedicated RX/token cores)`: Pins dedicated CPU cores for NIC receive and tokenization tasks to avoid scheduler overhead.

- `PIN(mem := superpages/pinned pools)`: Pins memory (e.g., hugepages) so buffers remain resident in RAM and are safe for zero-copy access.

- `INITNIC(RX queues bound to pinned mbufs)`: Initializes the NIC with RX queues backed by pinned mbuf pools.

- `ALLOCPINNEDMEMZONE`: Allocates a pinned memory zone $mz$ to store token IDs for direct Python consumption.

- `TOKENIZE($p_i \rightarrow mz$)`: Tokenizes packet $p_i$, writing token IDs into the pinned buffer $mz$ and returning the valid token-ID range $T_{\text{range}}$.

- `PUBLISHTOPYTHON($mz$, $T_{\text{range}}$)`: Publishes a zero-copy view of token IDs in $mz$ to Python for immediate use.

- `RECYCLERXBUF($p_i$)`: Recycles the RX packet buffer $p_i$ back to the mbuf pool for reuse.

## D STANDALONE TOKENIZER RESULTS

This section reports the full numeric results complementing the figures in the main paper. We provide throughput values (mean $\pm$ std over 20 trials) and tokenization latencies (average, median, 90th, and 99th percentiles) across three datasets: OpenWebText, GitHub Code, and Multilingual mC4.

Table 4: Throughput on OpenWebText (tokens/s); mean $\pm$ std over 20 trials.

| Tokens | PinTok | Rust | Python |
|---|---|---|---|
| 128 | $2,184,300 \pm 353,365$ | $226,336 \pm 11,358$ | $100,009 \pm 3,208$ |
| 256 | $2,359,882 \pm 242,993$ | $358,840 \pm 19,385$ | $146,068 \pm 6,997$ |
| 512 | $1,888,184 \pm 269,203$ | $559,276 \pm 24,345$ | $152,204 \pm 4,371$ |
| 1024 | $2,574,805 \pm 301,764$ | $964,473 \pm 68,975$ | $222,728 \pm 4,372$ |
| 2048 | $3,301,043 \pm 177,819$ | $1,978,916 \pm 132,838$ | $279,221 \pm 9,422$ |

Table 5: Throughput on GitHub Code (tokens/s); mean $\pm$ std over 20 trials.

| Tokens | PinTok | Rust | Python |
|---|---|---|---|
| 128 | $2,300,917 \pm 621,243$ | $289,783 \pm 16,447$ | $121,913 \pm 3,728$ |
| 256 | $3,184,080 \pm 552,462$ | $343,288 \pm 10,827$ | $136,032 \pm 2,729$ |
| 512 | $2,773,865 \pm 495,923$ | $611,789 \pm 23,261$ | $178,328 \pm 4,747$ |
| 1024 | $2,489,909 \pm 396,499$ | $735,289 \pm 73,780$ | $189,469 \pm 3,269$ |
| 2048 | $3,207,417 \pm 162,651$ | $1,287,249 \pm 178,258$ | $259,777 \pm 5,739$ |

Table 6: Throughput on Multilingual mC4 sample (tokens/s); mean $\pm$ std over 20 trials.

| Tokens | PinTok | Rust | Python |
|---|---|---|---|
| 128 | $1,780,746 \pm 339,897$ | $258,320 \pm 13,596$ | $101,254 \pm 2,703$ |
| 256 | $2,422,866 \pm 225,410$ | $305,435 \pm 12,758$ | $113,743 \pm 4,172$ |
| 512 | $1,683,989 \pm 260,929$ | $613,960 \pm 20,371$ | $168,092 \pm 4,073$ |
| 1024 | $1,863,004 \pm 122,291$ | $733,934 \pm 87,706$ | $172,058 \pm 2,909$ |
| 2048 | $2,364,459 \pm 111,131$ | $1,248,697 \pm 120,072$ | $232,933 \pm 4,754$ |

Table 7: Tokenization latency on OpenWebText (µs); mean over 20 trials. Lower is better.

| Tokens | PinTok | | | | Rust | | | | Python | | | |
|---|---|---|---|---|---|---|---|---|---|---|---|---|
| | Avg | P50 | P90 | P99 | Avg | P50 | P90 | P99 | Avg | P50 | P90 | P99 |
| 128 | 59 | 36 | 117 | 325 | 566 | 509 | 769 | 1,121 | 1,280 | 1,135 | 1,888 | 2,412 |
| 256 | 108 | 95 | 193 | 542 | 713 | 802 | 1,180 | 1,776 | 1,753 | 1,756 | 3,222 | 4,333 |
| 512 | 271 | 216 | 366 | 1,650 | 915 | 846 | 1,182 | 1,664 | 3,364 | 2,596 | 6,077 | 7,836 |
| 1024 | 398 | 328 | 577 | 2,184 | 1,062 | 912 | 1,307 | 4,011 | 4,598 | 2,815 | 10,037 | 12,346 |
| 2048 | 620 | 462 | 926 | 2,664 | 1,035 | 899 | 1,309 | 3,170 | 7,335 | 2,936 | 18,269 | 22,073 |

## E END-TO-END EMBEDDING PIPELINE RESULTS

This section presents the embedding throughput and latency results across three representative models—`e5-base-v2`, `ModernBERT-base`, and `embeddinggemma-300m`—on the OpenWebText dataset. The results highlight both average performance and distributional latency metrics to capture system efficiency under different token sizes.

Table 8: Tokenization latency on GitHub Code (µs); mean over 20 trials. Lower is better.

| Tokens | PinTok | | | | Rust | | | | Python | | | |
|---|---|---|---|---|---|---|---|---|---|---|---|---|
| | Avg | P50 | P90 | P99 | Avg | P50 | P90 | P99 | Avg | P50 | P90 | P99 |
| 128 | 56 | 28 | 106 | 473 | 442 | 394 | 618 | 891 | 1,050 | 903 | 1,583 | 2,186 |
| 256 | 80 | 50 | 153 | 458 | 746 | 686 | 986 | 1,428 | 1,882 | 1,689 | 2,693 | 3,486 |
| 512 | 185 | 168 | 287 | 1,011 | 837 | 1,022 | 1,477 | 1,984 | 2,871 | 3,718 | 5,111 | 6,376 |
| 1024 | 411 | 345 | 510 | 2,221 | 1,393 | 1,152 | 1,668 | 6,233 | 5,405 | 4,820 | 8,381 | 10,327 |
| 2048 | 639 | 606 | 841 | 1,950 | 1,591 | 1,184 | 1,919 | 8,316 | 7,884 | 6,253 | 13,617 | 16,175 |

Table 9: Tokenization latency on Multilingual (mC4 sample; µs); mean over 20 trials. Lower is better.

| Tokens | PinTok | | | | Rust | | | | Python | | | |
|---|---|---|---|---|---|---|---|---|---|---|---|---|
| | Avg | P50 | P90 | P99 | Avg | P50 | P90 | P99 | Avg | P50 | P90 | P99 |
| 128 | 72 | 47 | 140 | 612 | 496 | 440 | 689 | 1,121 | 1,264 | 1,084 | 2,026 | 2,567 |
| 256 | 106 | 79 | 212 | 446 | 838 | 784 | 1,115 | 1,601 | 2,251 | 2,142 | 3,441 | 4,256 |
| 512 | 304 | 272 | 413 | 1,630 | 834 | 946 | 1,488 | 2,022 | 3,046 | 3,653 | 5,532 | 8,511 |
| 1024 | 550 | 493 | 689 | 1,546 | 1,395 | 1,155 | 1,579 | 6,108 | 5,951 | 5,692 | 8,566 | 11,485 |
| 2048 | 866 | 876 | 1,187 | 2,386 | 1,640 | 1,181 | 1,731 | 9,535 | 8,792 | 6,253 | 15,316 | 18,975 |

Table 10: Embedding throughput on OpenWebText (tokens/s) with `e5-base-v2`; mean ± std over 5 trials.

| Tokens | PinTok | Rust | Python |
|---|---|---|---|
| 384 | 37,706.43 ± 678.93 | 27,289.27 ± 260.21 | 22,827.46 ± 166.07 |
| 448 | 44,388.80 ± 645.82 | 30,598.06 ± 407.97 | 25,391.84 ± 155.67 |
| 512 | 52,686.38 ± 847.33 | 33,941.27 ± 74.61 | 27,562.44 ± 299.99 |

Table 11: Embedding throughput on OpenWebText (tokens/s) with `ModernBERT-base`; mean ± std over 5 trials.

| Tokens | PinTok | Rust | Python |
|---|---|---|---|
| 512 | 20,505.75 ± 143.75 | 16,387.15 ± 101.70 | 14,908.04 ± 367.73 |
| 768 | 29,680.00 ± 660.51 | 21,956.97 ± 236.09 | 19,752.79 ± 346.90 |
| 1024 | 37,855.23 ± 199.71 | 26,135.28 ± 406.14 | 23,367.15 ± 401.33 |

Table 12: Embedding throughput on OpenWebText (tokens/s) with `embeddinggemma-300m`; mean ± std over 5 trials.

| Tokens | PinTok | Rust | Python |
|---|---|---|---|
| 512 | 9,709.27 ± 82.91 | 8,871.95 ± 69.38 | 8,358.14 ± 53.98 |
| 768 | 14,382.93 ± 119.59 | 12,515.31 ± 67.31 | 11,680.84 ± 41.30 |
| 1024 | 19,010.85 ± 99.77 | 15,963.13 ± 161.73 | 14,716.64 ± 40.86 |

Table 13: Embedding pipeline latency on OpenWebText with `e5-base-v2` (µs); mean over 5 trials. Lower is better.

| Tokens | PinTok | | | | Rust | | | | Python | | | |
|---|---|---|---|---|---|---|---|---|---|---|---|---|
| | Avg | P50 | P90 | P99 | Avg | P50 | P90 | P99 | Avg | P50 | P90 | P99 |
| 384 | 10,187 | 9,959 | 11,021 | 13,634 | 14,072 | 13,824 | 15,591 | 21,063 | 16,823 | 16,490 | 18,511 | 25,523 |
| 448 | 10,094 | 9,892 | 10,944 | 13,372 | 14,644 | 14,353 | 16,298 | 20,879 | 17,644 | 17,401 | 19,377 | 22,522 |
| 512 | 9,720 | 9,546 | 10,419 | 12,634 | 15,085 | 14,891 | 16,839 | 19,984 | 18,578 | 18,260 | 20,617 | 24,360 |

Table 14: Embedding pipeline latency on OpenWebText with `ModernBERT-base` (µs); mean over 5 trials. Lower is better.

| Tokens | PinTok | | | | Rust | | | | Python | | | |
|---|---|---|---|---|---|---|---|---|---|---|---|---|
| | Avg | P50 | P90 | P99 | Avg | P50 | P90 | P99 | Avg | P50 | P90 | P99 |
| 512 | 24,970 | 24,324 | 27,178 | 33,261 | 31,245 | 30,500 | 34,493 | 40,895 | 34,361 | 33,774 | 37,863 | 43,621 |
| 768 | 25,886 | 25,269 | 28,254 | 34,897 | 34,981 | 34,216 | 38,555 | 47,606 | 38,890 | 38,285 | 42,981 | 49,534 |
| 1024 | 27,051 | 26,247 | 29,600 | 36,358 | 39,188 | 38,462 | 43,713 | 49,983 | 43,833 | 43,239 | 48,914 | 55,078 |

Table 15: Embedding pipeline latency on OpenWebText with `embeddinggemma-300m` (µs); mean over 5 trials. Lower is better.

| Tokens | PinTok | | | | Rust | | | | Python | | | |
|---|---|---|---|---|---|---|---|---|---|---|---|---|
| | Avg | P50 | P90 | P99 | Avg | P50 | P90 | P99 | Avg | P50 | P90 | P99 |
| 512 | 52,736 | 51,533 | 57,201 | 62,566 | 57,713 | 56,787 | 62,666 | 68,547 | 61,260 | 60,166 | 66,371 | 73,348 |
| 768 | 53,400 | 52,064 | 57,926 | 63,483 | 61,366 | 60,242 | 66,709 | 73,986 | 65,749 | 64,544 | 71,485 | 81,191 |
| 1024 | 53,865 | 52,361 | 58,296 | 64,564 | 64,153 | 63,004 | 69,433 | 77,639 | 69,582 | 68,529 | 75,400 | 83,681 |

## F BYTE PAIR ENCODING (BPE) ALGORITHM

The BPE algorithm that we have utilized for the evaluations has two parts: training and inference, i.e., tokenization. Although the focus of this work is mainly on inference, we are sharing both the training and inference logic for reference. Furthermore, each model has model-specific training and inference logic that adds custom logic to the provided code, but the high-level idea stays the same.

### PHASE 1: TRAINING (LEARNING MERGES)

This part of the algorithm takes as input a text corpus and a desired vocabulary size. It iteratively merges the most frequent adjacent pairs of symbols to build a vocabulary and a list of merge rules.

```
function train_bpe(corpus, vocab_size)
    % 1. Initialization
    vocab ← set of all unique characters in corpus
    pre_tokenized_corpus ← split corpus by whitespace, then split each
    word into characters
    num_merges ← vocab_size - size(vocab)
    merge_rules ← empty list

    % 2. Iterative Merging
    for i = 1 to num_merges do
        % Get statistics of all adjacent pairs
        pair_counts ← get_pair_stats(pre_tokenized_corpus)

        % Find the most frequent pair
        if pair_counts is empty then
            break % No more pairs to merge
        end if
        best_pair ← argmax(pair_counts) % e.g., ('t', 'h')
```

```
        % Create the new token by merging the best pair
        new_token ← best_pair[0] + best_pair[1] % e.g., "th"

        % Add the new token and merge rule
        add new_token to vocab
        add best_pair to merge_rules (in order of merging)

        % Update the corpus by replacing all occurrences of `best_pair`
        pre_tokenized_corpus ← merge_pair(pre_tokenized_corpus, best_pair,
    new_token)
    end for

    return vocab, merge_rules
end function

% Helper function to compute pair status
function get_pair_stats(corpus)
    counts ← empty dictionary
    for each word in corpus do
        symbols ← tokens in the word
        for j = 0 to size(symbols) - 2 do
            pair ← (symbols[j], symbols[j+1])
            increment count for pair in counts
        end for
    end for
    return counts
end function

function merge_pair(corpus, pair_to_merge, new_token)
    new_corpus ← empty list
    for each word in corpus do
        new_word ← replace all occurrences of pair_to_merge in word with
    new_token
        add new_word to new_corpus
    end for
    return new_corpus
end function
```

Pseudocode for BPE Training to build the dictionaries

PHASE 2: INFERENCE / TOKENIZATION

Given a query and the learned merge rules from the training step, this function tokenizes the text by applying the rules in the same order that they were learned.

```
function bpe_tokenize(text, merge_rules)
    % 1. Pre-tokenization
    words ← split text by whitespace
    tokenized_output ← empty list

    % 2. Apply merge rules to each word
    for each word in words do
        % Start with individual characters
        word_tokens ← split word into characters

        % Iteratively apply all learned merge rules
        for each pair in merge_rules do
            new_word_tokens ← empty list
            i ← 0
            while i < size(word_tokens) do
                if i < size(word_tokens) - 1 and (word_tokens[i],
    word_tokens[i+1]) == pair then
```

```
                        add (word_tokens[i] + word_tokens[i+1]) to
    new_word_tokens
                        i ← i + 2
                else
                    add word_tokens[i] to new_word_tokens
                    i ← i + 1
                end if
            end while
            word_tokens ← new_word_tokens
        end for

        add word_tokens to tokenized_output
    end for

    return flatten(tokenized_output)
end function
```

Pseudocode for BPE at Tokenization

