# OpenReview forum: "PinTok: Tokenizers Deserve Dedicated Pinned CPU-Compute and Memory"
_ICLR.cc/2026/Conference — Submitted to ICLR 2026_

### Official Review · Reviewer_XPZb · 2025-10-25

**Soundness:** 3
**Presentation:** 2
**Contribution:** 3
**Rating:** 4
**Confidence:** 4

**Summary:**

This paper presents PinTok, a software implementation of tokenization that improves upon standard implementations in terms of throughput and latency. These improvements stem from three optimizations: (1) pinning the CPU cores and RAM used by the tokenizer; (2) avoiding process scheduling costs and context switching by continuously polling for incoming inputs to tokenize from the network card, which encourages the OS to keep it on the same core; and (3) copying incoming inputs directly from the network card into userspace RAM, bypassing kernel space RAM. The results in the paper show that PinTok has much better throughput and latency on a variety of benchmarks versus Rust and Python implementations of BPE, and the proportion of runtime that is reduced improves for smaller models.

**Strengths:**

The paper's goal is to provide a faster tokenizer implementation, and the results show nice improvements. Tokenizers are used wherever LLMs are used, so a speedup to the tokenizer could be a significant contribution. I'm not aware of prior work that has applied these techniques to tokenizers. The work generally appears to be of good quality, with a reasonable breadth of models and datasets in the experimental section. Section 2 is helpful for providing technical background and breaking down the potential areas for speedups. The authors helpfully quantify what proportion of time is taken up by the tokenizer in a small-scale LM setup, and therefore the headroom for speeding up the tokenizer.

**Weaknesses:**

A lof of my issues with this paper have to do with clarity; I have included a number of clarifying questions in the Questions section.

1. The techniques that PinTok relies upon have tradeoffs; otherwise, why couldn't we use them for every aspect of the LM pipeline, instead of just the tokenizer? For instance, pinning the tokenizer to a CPU core obviously takes that resource away from all other processes running on the system, potentially slowing down other aspects of the LM pipeline, or other processes that need to run on the same system. The experimental results show that the overall pipeline is still faster, but I would like to see more discussion of the tradeoffs of each of these techniques, how PinTok is able to get away with using these tricks despite the tradeoffs, and why the benefits outweigh the costs specifically in the case of tokenization, as opposed to any other aspect of the LM pipeline.
1. It isn't clear why the NIC comes into play. What I've inferred from reading through the paper is that a process must use PinTok by having PinTok run in its own process and must use the NIC as an IPC mechanism. On this point, wouldn't it be better to avoid IPC altogether and have the code that runs the LM and the code that tokenizes run in one process that's pinned to a dedicated CPU core? This would eliminate all of the overhead from using the NIC and from polling.
1. The polling trick seems like an undesirable solution to avoiding context switching, as it's essentially a busy-waiting loop, and it doesn't guarantee that context switches won't occur. How often does PinTok poll for inputs?
1. I found Algorithms 1 and 2 to be more cryptic than helpful, since the reader needs to infer a lot of details from the names of the functions.
1. Fig 5: I don't understand what the x axis is. Is each point along the x axis measuring the speed of tokenizing a single input of a certain length? If not, how many different inputs were used? How was the input string selected?
1. If I understand correctly, the authors use Rust and Python reimplementations of BPE as their baselines. I would rather they compare to existing libraries, namely Hugging Face, SentencePiece, and tiktoken. Otherwise, it's not clear if the speedups are actually attributable to their reimplementation.
1. It should also be noted that the primary contribution of this paper is a software artifact, but the authors have not provided any code.
1. 012: "yet it has not traditionally been viewed as a component of LLMs worth accelerating" -- I don't think this is true. Consider the C++ implementation of SentencePiece or the Rust implementation in Hugging Face. See also https://aclanthology.org/2023.findings-acl.38/.

**Questions:**

1. Does PinTok also take care of pretokenization (splitting on whitespace and punctuation, grouping digits, etc.), or just the BPE algorithm proper?
1. Did you carefully unit-test your reimplementation of the tokenizer to ensure that it is equivalent to the existing implementations in Hugging Face and SentencePiece?
1. 477: It sounds like because you're chunking up the input into packets and then tokenizing, you're not producing a result that's equivalent to the original implementation. Is this the case?
1. 367: Do all models in this paper use BPE? What do you mean by different tokenization algorithms?
1. What language is PinTok written in?
1. Table 3: How is latency saving computed?
1. Can you measure the actual number of context switches? Does it decrease with PinTok?
1. Can you actually quantify where the speedups come from, following the equations in Section 2? Which optimizations are the most important?

Typos:
* 069: works -> work
* There are a few \citep vs. \citet mistakes
* 083: ungrammatical
* 100: ungrammatical
* 191: minimizer -> minimize
* Fig 5: plot titles overlap
* 436: Appendix XXX
* Why is Time to First Byte abbreviated as TTFT?

---

> ### Author Response · Authors · 2025-11-21
>
> We are happy to see the reviewer recognizing both the novelty and the significance of our contribution. We respond to the individual concerns below.
>
> **Discussion of tradeoffs**: Please refer to the common response. We also clarify that tokenization is the only major component in the LLM pipeline that is not GPU-accelerated, and the techniques used in PinTok do not apply to the GPU-accelerated components.
>
> **why the NIC comes into play**:
> The NIC must have a compatible driver to allow DPDK to map the NIC's hardware into user space, which enables features like polling for packets instead of using interrupts.
> Most modern NICs are DPDK-compatible.
>
> **polling trick seems undesirable**: Polling for packet is a widely used technique in high-performance networking to achieve low-latency and minimal context switching.
> For example, companies like Nvidia widely uses DPDK and poll-mode drivers already for their inference system
> (https://developer.nvidia.com/networking/dpdk).
>
> **Algorithms 1 and 2 are cryptic**: Thank you for this feedback. In the appendix of our updated manuscript, we provide further details explaining the helper functions used in Algorithms 1 and 2 to aid in understanding of the algorithms.
>
> **Fig 5 clarification**: The x-axis for Fig. 5 represents the input length for each query. We have updated the figure and the description to reduce confusion. Thank you for your comment.
>
> **Rust and Python BPE as baselines**: To clarify, we have `not' used reimplementatino of Rust and Python BPE algorithms. Our baseline is obtained from using the unaltered Huggingface Tokenizer implentations.
>
> **Code isn't available**: Please refer to the common response. The code is now available.
>
> **tokenization has not traditionally been viewed as a component of LLMs worth accelerating is not true**: Thank you for the thoughtful point. We have adjusted the claim to be more factually accurate in our updated manuscript.
>
> **PinTok also take care of pretokenization?**: Thank you for the thoughtful question. Yes, PinTok includes pretokenization as well, and our measurements consider the entire tokenization process, not just the BPE algorithm.
>
>
> **unit-test PinTok to ensure it is equivalent?**: Yes, to verify the implementations, we have careful parity checked implemented across multiple test queries.
> There are open questions about chunking, which we addressed in the bullet below.
>
>
>
> **Concerns about Chunking**: As mentioned in the conclusion section, concerns regarding chunking at packet boundary are subtle corner cases with little performance potential impact.
> For our experiments, we have not seen this issue since we focus on models with small context sizes, thus the inputs are smaller than a single packet size.
> We plan to target this with a more robust tokenization algorithm in the future work.
>
> **Do all models in this paper use BPE?**: Yes, we used BPE for all our experiments to keep the latency comparisons consistent. However, our proposed methodology is tokenizer-algorithm-agnostic; the core concepts can be applied to any type of tokenizer. Evaluating additional tokenization algorithms on top of the PinTok framework is an important direction for future work.
>
>
> **What language is PinTok written in?**: PinTok is primarily written in C with Python interfaces to easily be used with existing Python scripts. Again, code is now available as supplementary material.
>
>
>
> **Table 3: How is latency saving computed?** Latency savings (Column 5) in Table 3 are computed by directly comparing the end-to-end latencies of PinTok and the Python tokenizer when coupled with each application in Column 1. The key point is that PinTok provides the largest benefits when paired with smaller models.
>
> **Can you measure the actual number of context switches?**: Measuring the exact number of context switches for only the tokenizer is difficult, but it is still valuable to approximate it by comparing the full Python experiment script execution. Using ten queries of length 2048 sampled from the `openwebtext` dataset, we observe 612 context switches for PinTok and 3,043 for the Python tokenizer. These counts include all experiment-script overhead, but the difference arises solely from the tokenizer implementations.
>
>
> **Ablating many the sources of speedup**: We can break down the speedup into two measurable components: extraction (packet to text) and the tokenization algorithm itself. Using 2048-token samples from the `openwebtext` dataset, PinTok shows 809.21 µs extraction time and 2,213.96 µs tokenization time, while the Python tokenizer shows 12,079.59 µs for extraction and 30,749.62 µs for tokenization. This indicates that most of the speedup comes from the tokenization algorithm.
>
>
> **Typos and grammatical issues**: Thank you for pointing these out. We have addressed them in the updated manuscript.
>
>
> $$ $$
>
> Given that we positively address the main concerns of the reviewer, we kindly ask the reviewer to consider raising the score.

---

### Official Review · Reviewer_9a6d · 2025-10-30

**Soundness:** 3
**Presentation:** 4
**Contribution:** 4
**Rating:** 8
**Confidence:** 4

**Summary:**

The authors present Pinned Tokenizer (PinTok) for addressing scheduling delays, core selection,
data copying, and other system-level costs that standard LLM tokenizers face when working in latency-sensitive applications. Tokenizers translate strings inputs into model ingestible representations, but despite this importance or often just implemented as dictionary lookups. PinTok improves their implementation specifically by introducing techniques for core and memory pinning, scheduling and context switch avoidance, and processing avoidance. Using PiTok shows latency reductions reaching 95%, on average, and throughput increases of up to 2000%.

**Strengths:**

- The paper is, as far as I'm aware, a first to take a systems level treatment for optimizing tokenization. This is quite novel, and differs from the traditional focus of algorithmic improvement.
- The approach is "agnostic to the specific tokenizer algorithm", making it a drop-in replacement that has substantial performance upsides. Moreover, the approach doesn't depend on specific hardware, making it broadly applicable.
- The problem is well motivated. Specifically, in Table 1 they show significant proportion of time is spent on tokenization for smaller models. Likewise in Table 2 the authors clearly show the major sources of latency during tokenization.
- The experimental setup is thorough. The authors benchmark against web text, code, and multilingual datasets — representing the diverse sources of information that is commonly tokenized for LLMs. The number of trials for each benchmark give good evidence that the approach is generalizable.
- The presentation of the paper is well done. The problem and solution are clear and the experiments have a clear focus and takeaway. The writing is easy to follow.

**Weaknesses:**

- Line 436: Missing the link ("Appendix XXX") to detailed numerical results for all datasets and configurations.
- Unclear what the trade-off is with this approach. Specifically, the dedicates resources it requires to pin CPU cores and 8GB+ of pages exclusively for tokenization — is this too significant in some setups?
- No statistical tests of significant used, however the experimental setup is rather robust based the number of repitions they performed and the range in experiments.
- The major performance improvements occur in specific environments. Table 2 shows that PinTok is only slightly faster than Rust and Python tokenizers for  `GPT-OSS 120b query (TTFT)` and `Qwen3 4B query (TTFT)`. And the performances are generally smaller for larger models which are likely to most benefit do to their large scales.
- The authors describe the many different sources of latency, however the experimental setup doesn't do any sort of ablation study to show which of their techniques contributes the most to the performance improvements (e.g., pinning versus the page size).
- Reproducibility concerns: the code isn't available with the paper, but may be later.

**Questions:**

- Is Appendix C "WHY EVEN CONSIDER USING PYTHON TOKENIZERS?" necessary? This thought never crossed my mind and it seems like python tokenizers should be the baseline due to their widespread usage.
- Is there any reason to believe that this approach wouldn't result in a fully accurate and equivalent tokenization as the baselines? Did you perform any checks to validate that the results match baselines exactly?

---

> ### Author Response · Authors · 2025-11-21
>
> We are very pleased that the reviewer found our work novel and the gains substantial. We respond to the individual concerns below.
>
>
> **"Appendix XXX" typo**: Thank you! We have fixed the reference in the updated manuscript.
>
> **trade-off is with this approach**: Please refer to the common response.
>
>
> **No statistical tests of significant used, although the experimental setup is robust**:
> As the reviewer notes, we believe the consistent results across a wide range of settings sufficiently demonstrate the speedup, even without a formal statistical test or reported p-value.
>
> **major improvements occur in specific environments**: Please refer to the common response.
>
> **Ablating many different sources of latency**: Decomposing latency exactly as in the equations is difficult, but we can break down the speedup into two measurable components: extraction (packet to text) and the tokenization algorithm itself. Using 2048-token samples from the `openwebtext` dataset, PinTok shows 809.21 µs extraction time and 2,213.96 µs tokenization time, while the Python tokenizer shows 12,079.59 µs for extraction and 30,749.62 µs for tokenization. This indicates that most of the speedup comes from the tokenization algorithm rather than the preprocessing step.
>
>
> **Reproducibility, code isn't available**: Please refer to the common response. The code is now available.
>
>
> **Is Appendix C necessary?**: Thank you for this point. In hindsight, we agree with the reviewer. We have removed this appendix in the updated manuscript.
>
>
> **Is PinTok tokenization equivalent?**:
> A case where there may be discrepancies between PinTok and the traditional tokenizer is when token boundaries cross packet boundaries.
> As mentioned in the conclusion section, while we believe that such cases are subtle corner cases with little performance impact, we plan to target this in future work.
> For our experiments, we have not seen this issue since we focus on models with small context sizes, thus the inputs are smaller than a single packet size.

---

### Official Review · Reviewer_sqje · 2025-11-10

**Soundness:** 2
**Presentation:** 2
**Contribution:** 1
**Rating:** 2
**Confidence:** 4

**Summary:**

The system PinTok aims to build a system to speedup the tokenization bottleneck in many system.  They attempt to reduce the cost via core and memory pinning, network packet dedup, and reduce context switching overhead.

**Strengths:**

In practice, I’ve seen the cost of tokenizers. I find this work to be a valuable contribution in terms of reducing the overall cost of tokenization time.

**Weaknesses:**

1. I find the systems contribution to be slightly weak/lacking enough new novel approaches. The core idea boils down to just pinning/huge pages. The projects feels like it can do a lot more work/and is incomplete.
2. The optimizations provided are not very specific to tokenizer handling and seem to not take advantage of tokenizer properties.

**Questions:**

Can you describe what makes the Pintok system challenging? Why are these optimizations specific to tokenization?

---

> ### Author Response · Authors · 2025-11-21
>
> We respectfully disagree with the reviewer’s assessment that the systems contribution is lacking.
>
> The primary contribution of our work is the tokenizer code itself, which represents a substantial engineering effort (~20,000 lines low-level C code with Python interfaces). In terms of solving systems-level challenges, one is performing tokenization directly in the network packet-processing path to eliminate redundant processing that typically occurs multiple times when receiving data. Because network protocols and programs are heavily optimized for integer computation and memory lookups, and because tokenizers also mainly rely on these operations, PinTok's approach is particularly effective and novel.
>
> Further, our code is structured to be conveniently used with PyTorch in a drop-in manner. With only a few minutes of installation time, there is essentially no downside to using PinTok instead of the default (usually the Huggingface tokenizers). In settings where tokenizer overhead is small, no harm is done; in the many settings where tokenization constitutes a meaningful fraction of runtime, the user benefits from the speedups. If our tokenizer sees moderate adoption, the aggregate benefit to the community will be large (and not to mention that the individual benefits will be large for the use cases highlighted in the common response).
>
> In summary, our paper presents a non-trivial systems engineering contribution with the potential to provide substantial value to the ICLR community. We respectfully ask the reviewer to take this into consideration in assessing the value that our work provides.

---

### Official Review · Reviewer_7Nnk · 2025-11-10

**Soundness:** 2
**Presentation:** 3
**Contribution:** 2
**Rating:** 4
**Confidence:** 3

**Summary:**

This paper introduces PinTok, which is a tokenizer implementation that uses core and memory pinning, while avoiding duplicate network packet copying & processing as well as context switching & scheduling. The paper presents basics of the infrastructure design space and timing considerations along with the method. PinTok provides massive speedups in the experimental settings that are presented.

**Strengths:**

The paper has several merits:

* **Clarity**. The paper clearly presents the PinTok method. Presents hardware / os /networking concepts in a way that can be easily understood by ICLR community.
* **Effectiveness**. The empirical setting compares the throughput/latency of standard tokenizers to PinTok. There is significant speedups due to PinTok's dedicated design. This is impressive and something that can likely be of interest to many working on models/applications in which tokenization takes up significant time.

**Weaknesses:**

My concerns about the paper are:

* **Details about Tokenize -> Forward Pass**- Suppose we are receiving data continuously, tokenizing it, pushing it through the model (at least forward pass), why can't we do pipeline-parallelism to have the CPUs of the machine be processing & tokenizing data of batch i+1 and the accelerators running the model on batch i? How does this setting relate to the wall clock %s that are shown on Table 1? I am a bit confused because it seems like this can hide a large part of the costs of tokenization for many such applications? (Apologies if I have missed this or if there is something obvious here that I did not understand. However, I looked carefully again at the paper am still confused hence raising this issue.)
* **Scaling questions** - The paper does not, as far as I can tell, report how the tokenization performance scales with number of cores / parallelism? This seems important not only to understand if there are settings where we need to parallelize differently with the proposed method. Furthermore, it would be interesting to understand if things like vocab size plays a roll in the efficiency of the method. These seem like standard considerations in such a setting about scalable tokenization.

Minor:
"Line 436 Appendix XXX"

**Questions:**

Please see weakness above. In summary:
1) Can we tokenize batch i+1 while batch i is processed with the transformer?
2) Does the approach scale linearly with number of CPUs?
3) Does increasing vocab size affect PinTok differently than normal tokenizers?
4) When would PinTok not be the right tokenizer implementation to use?

---

> ### Author Response · Authors · 2025-11-21
>
> We are happy to hear that the reviewer found the speedups brought by PinTok to be impressive. We respond to the individual concerns in the following:
>
>
> **Hiding cost of tokenization when GPU load is continual**:
> We first refer the reviewer to Point \#2 in the common response, which outlines the setup in which PinTok is expected to deliver meaningful speedup benefits.
>
> Indeed, the reviewer is correct that when a continual GPU load is provided, tokenization cost can be completely hidden, and in such scenarios, the tokenizer’s efficiency does not affect overall throughput.
> However, in many of the applications we target, latency, specifically Time to First Token (TTFT), is the critical metric. In these settings, tokenization cost cannot be hidden and PinTok's benefit directly improves the user-perceived latency.
>
>
> **Scalability with CPUs**:
> Current tokenization methods are inherently sequential, so neither PinTok nor any other existing tokenizer can scale its per-sequence latency with more CPU cores. What does scale linearly is the ability to process multiple independent queries in parallel, each on its own CPU core. Thus, multi-core scaling helps throughput.
>
>
> **Scalability with vocab size**: PinTok is less sensitive to increases in vocabulary size because it stores its data in a single large page, minimizing the need to use multiple pages. In contrast, conventional tokenizers distribute vocabulary data across many standard memory pages, so a larger vocabulary typically causes more page misses and degraded memory locality. PinTok’s memory layout avoids this issue and maintains more stable performance as the vocabulary size grows.
>
> **When would PinTok not be the right tokenizer?**:
> If the server is running other CPU-intensive workloads (outside normal LLM inference or training), dedicating a pinned CPU core to PinTok may not be optimal. However, such environments are uncommon in typical language model deployments as it negatively impacts the main goal of such environment, which is to have high performing language models.
>
>
> **"Appendix XXX" typo**: Thank you! We have fixed the reference in the updated manuscript.
>
> $$ $$
>
> Given that we positively address main concerns of the reviewer, we kindly ask the reviewer to consider raising the score.

---

### Author Response · Authors · 2025-11-21
**Common response**

We thank the reviewers for their detailed and constructive feedback. The reviewers generally found our problem formulation well-motivated, the speedups significant, and the contribution valuable. The reviewers raised several shared concerns, which we address in our common response below.

**General concern 1: Availability of code.**
The reviewers raise the excellent point that our primary contribution is the software artifact. We have now made our full implementation available as supplementary material.


**General concern 2: Applications that would benefit from PinTok's acceleration**


We expect PinTok to provide the greatest benefit in applications that rely on ensembles of small language models, such as embedding models, vector databases for Retrieval-Augmented Generation (RAG), and agentic AI systems, where tokenizer latency constitutes a more significant portion of end-to-end runtime. As the compute cost of LLMs continues to fall (with some estimates suggesting a 40x reduction per year), these use cases will become more prevalent.



**General concern 3: Is there a tradeoff to using CPU resources?**
In typical LLM-inference or training pipelines, CPUs are often significantly underutilized. In such settings, dedicating a single pinned CPU core incurs no practical cost and does not interfere with other processes. In cases where CPUs are highly utilized for tokenization, CPUs are effectively being pinned for tokenization anyways. Consequently, there is effectively no tradeoff: PinTok simply leverages otherwise idle CPU capacity to achieve strictly superior efficiency.

$$ $$

We address the reviewers' individual concerns in the corresponding individual responses.

---

### Author Response · Authors · 2025-12-02
**Quick Message to the New AC**

We thank the new AC for taking on the difficult task of making final decisions without the full reviewer-author interaction data. As the discussion period was suspended before any of the reviewers could respond, we would like to provide our interpretation of the rebuttal.


**Common Response.**
The reviewers had some shared points, which we addressed in our common response. We believe those issues were resolved convincingly.

**Reviewer 9a6d.**
Reviewer 9a6d understood our work clearly and expressed strong appreciation. We reiterate Reviewer 9a6d's point that our work is "first to take a systems level treatment for optimizing tokenization. This is quite novel"

**Reviewer SQJE.**
Reviewer SQJE assigned a low score despite agreeing with the premise that accelerating tokenizers is valuable. Our understanding is that the reviewer considered our contribution insufficiently challenging. We firmly disagree with this notion: our work represents a significant engineering effort and several nontrivial systems-level insights throughout the design and implementation.

**Reviewers 7Nnk and XPZb.**
Both reviewers acknowledge that our work offers improvements that are “significant” and “of good quality.” The concerns they raise are not inherent to the system design itself, but instead relate to minor clarifications or updates to the paper, all of which we have successfully addressed.

---

### Meta-Review · Area_Chair_NKAs · 2026-01-06

**Summary:**

While one reviewer strongly supported this submission, the others raised significant concerns. Even after considering the authors' rebuttal, it remains difficult to definitively assess whether these core issues have been fully resolved.

**Reviewer Concerns:**

Remaining concerns:

1. Contribution/novelty might be limited.

2. The method is useful in some applications but will not improve some scenarios like continual GPU load.

**Reviewer Scores:**

8/6/4/2

---

### Decision · Program_Chairs · 2026-01-26

Reject